# Sustained and Targeted Delivery of Self-Assembled Doxorubicin Nonapeptides Using pH-Responsive Hydrogels for Osteosarcoma Chemotherapy

**DOI:** 10.3390/pharmaceutics15020668

**Published:** 2023-02-16

**Authors:** Jie Zhu, Rui Gao, Zhongshi Wang, Zhiming Cheng, Zhonghua Xu, Zaiyang Liu, Yiqun Wu, Min Wang, Yuan Zhang

**Affiliations:** 1Department of Neurology, Daping Hospital, Army Medical University, Chongqing 400038, China; 2State Key Laboratory of Natural Medicines, China Pharmaceutical University, Nanjing 210009, China; 3Joint Disease & Sport Medicine Center, Department of Orthopedics, Xinqiao Hospital, Army Medical University, Chongqing 400038, China

**Keywords:** peptide hydrogel, pH responsive, tumor targeted, drug delivery, doxorubicin, osteosarcoma

## Abstract

While chemotherapeutic agents have particularly potent effects in many types of cancer, their clinical applications are still far from satisfactory due to off-target drug exposure, chemotherapy resistance, and adverse effects, especially in osteosarcoma. Therefore, it is clinically promising to construct a novel tumor-targeted drug delivery system to control drug release and alleviate side effects. In this study, a pH-responsive nonapeptide hydrogel was designed and fabricated for the tumor-targeted drug delivery of doxorubicin (DOX). Using a solid-phase synthesis method, a nonapeptide named P1 peptide that is structurally akin to surfactant-like peptides (SLPs) due to its hydrophobic tail and hydrophilic head was synthesized. The physicochemical properties of the P1 hydrogel were characterized via encapsulation capacity, transmission electron microscopy (TEM), circular dichroism (CD), zeta potential, rheological analysis, and drug release studies. We also used in vitro and in vivo experiments to investigate the cytocompatibility and tumor inhibitory efficacy of the drug-loaded peptide hydrogel. The P1 peptide could self-assemble into biodegradable hydrogels under neutral conditions, and the prepared drug-loaded hydrogels exhibited good injectability and biocompatibility. The in vitro drug release studies showed that DOX-P1 hydrogels had high sensitivity to acidic conditions (pH 5.8 versus 7.4, up to 3.6-fold). Furthermore, the in vivo experiments demonstrated that the DOX-P1 hydrogel could not only amplify the therapeutic effect but also increase DOX accumulation at the tumor site. Our study proposes a promising approach to designing a pH-responsive hydrogel with controlled doxorubicin-release action based on self-assembled nonapeptides for targeted chemotherapy.

## 1. Introduction

Osteosarcoma is an extremely malignant bone neoplasm that arises from primitive transformed cells of mesenchymal origin, and it exhibits aggressive invasion properties [1]. Although its overall occurrence represents less than 1% of all cancers diagnosed annually in the United States and 3% of all childhood cancers, its irreversible results, such as amputation, early metastasis, and unsatisfactory 5-year survival rate (less than 50%), can be devastating to families and society [2]. Despite the development of new medical approaches in recent decades, chemotherapy using cytotoxic agents remains the main strategy for conservative treatment [3]. Doxorubicin (DOX) is a broad-spectrum anticancer agent that has a particularly potent effect in osteosarcoma [4,5]. However, its clinical application is still far from satisfactory, most likely due to off-target drug exposure, chemotherapy resistance, and severe side effects, such as cardiotoxicity [6] and bone marrow suppression [7]. Therefore, it is clinically appealing but challenging to construct a novel tumor-targeted drug delivery system of DOX to control drug release and alleviate side effects.

Hydrogels are three-dimensional (3D) porous networks with high water content formed through physical or chemical crosslinking [8], and have been envisioned as potential candidates in biomedical fields. While synthetic polymer hydrogels with excellent mechanical performance have been studied in recent years, their potential toxicity impedes their in vivo applications [9]. As a class of natural polymer hydrogels, peptide hydrogels have bright prospects in drug delivery systems owing to their outstanding biocompatibility, biodegradability, and structural diversity [10]. More importantly, peptide hydrogels with rational design can not only control drug release by varying the extent of crosslinking but also prolong drug retention at the tumor site via the spatial limit [11]. Moreover, peptide hydrogels can be designed as ‘smart’ materials [12]. They can trigger gel–sol or sol–gel transition in response to external stimuli, including pH [13], temperature [14], light irradiation [15], and ionic strength [16]. Compared with the surrounding normal tissues, the pH value of the tumor microenvironment was slightly lower in [17]. Therefore, this dynamic pH responsiveness of peptide hydrogels is of particular use in targeted cancer therapy [18].

Surfactant-like peptides (SLPs) are a kind of self-assembling peptide with a structure like that of traditional surfactants [19]. The amphiphilic structure of SLPs usually contains several consecutive hydrophobic amino acids for a single hydrophobic tail and one or two charged residues for a unique hydrophilic head [20]. Although they have strong self-assembly ability in aqueous solution, their poor drug-loading capacity and stability severely hinder their application in drug delivery [21]. To improve their performance and sensitivity to the tumor microenvironment, we herein designed and prepared a novel pH-responsive peptide hydrogel for the localized delivery of DOX. The sequence of this nonapeptide is Ac-FFFGSLKGK (named P1). 

In this work, consecutive-block FFF and charged-block KGK structurally mimicked the hydrophobic tail and hydrophilic head of SLPs, which are used to modulate molecular assembly and hydrogel formation [22]. We also designed Ac-FFFGSLKG (P0) and Ac-FFFGSLKGD (P2) for comparison. Under neutral conditions, P1 could self-assemble into a stable hydrogel without the addition of other additives. At the same time, DOX could be effectively encapsulated into the hydrogels. In the tumor microenvironment, the phase transition of DOX-P1 peptide hydrogels facilitated the sustained release of DOX. The properties of the P1 hydrogel were characterized via micromorphology, secondary structures, in vitro release studies, and rheology studies. We evaluated its antitumor efficacy in K12 osteosarcoma orthotopically grown in BALB/c mice. In both in vitro and in vivo studies, hydrogels could significantly retard tumor growth and alleviate side effects. Furthermore, they could also enhance drug retention in the tumor regions and reduce off-site distribution (Figure 1). In summary, the P1 hydrogel could amplify the anticancer efficacy of DOX, showing broad application potential as a local drug delivery platform. 

## 2. Materials and Methods

### 2.1. Materials

Standard 9-fluorenylmethoxycarbonyl (Fmoc)-protected amino acid, Ac-phenylalanine, N-hydroxy-benzotriazole (HOBt), Wang resin, and 4-dimethyl-aminopyridine (DMAP) were purchased from GL Biochem (Shanghai, China). Doxorubicin hydrochloride (DOX) was obtained from Aladdin (Shanghai, China). All other unspecified reagents were purchased from Sigma-Aldrich (St. Louis, MO, USA). Murine K12 osteosarcoma cells and normal NIH3T3 fibroblast cells were obtained from Keygen Biotech (Jiangsu, China) and cultured in a humidified incubator (37 °C, 5% CO_2_). Female BALB/c mice with K12 cells were supplied by the Qinglong Mountain Center (Jiangsu, China). All animal procedures were approved and supervised by the Animal Experiment Ethics Committee of the Army Medical University (Chongqing, China).

### 2.2. Methods

#### 2.2.1. Synthesis and Purification of the Peptide

Peptides were synthesized using a standard solid-phase synthesis (SPPS) method [23]. We utilized Wang resin for solid carriers as the terminal of P1 was carboxyl. After the crude products were cleaved from the resins, the purification of peptides was performed via Shimadzu high performance liquid chromatography (LC-8A, Tokyo, Japan) using a reversed-phase preparative C18 column (340 × 28 nm, 5 μm) and a binary mobile phase of water/acetonitrile (0.1% TFA). The molecular mass of the final products was verified via liquid chromatography–mass spectrometry (Thermo-fisher, New York, NY, USA).

#### 2.2.2. Preparation of Hydrogels

An appropriate amount of the peptide was dissolved in deionized water at room temperature. Then, the mixture solution was immediately vortexed for 10 s to ensure adequate and homogeneous dissolution. The pH was adjusted to 7.4 using 0.5 M NaOH. After 15 min, a stable hydrogel was formed using a simple vial inversion method. To prepare DOX-loaded peptide hydrogels, a certain amount of DOX was dissolved in deionized water following the steps outlined above.

#### 2.2.3. Gelation Behavior at Different pH Values

Several peptide solutions of different concentrations were prepared following the above method. The desired pH values (pH 5.8, 7.4, and 9.2) were adjusted through the addition of 0.5 M NaOH or HCl solutions. The phenomena of gel–sol and sol–gel were observed by everting the vials.

#### 2.2.4. Encapsulation Efficiency of the Peptide Hydrogel

To determine the encapsulation efficiency of DOX in different peptides, peptides (P1, P2) were dissolved in DOX solutions (1 mg/mL) to a concentration of 15 mg/mL and with the pH value adjusted to 7.4. The solutions were centrifuged at 1000 rpm for 2 min to remove bubbles and left overnight in a refrigerator at 4 °C. Then, 1000 μL phosphate buffer saline (pH 7.4) was used to rinse the surface of the hydrogels 3 times. All phosphate buffer saline solutions needed to be recovered prior to testing. The encapsulation efficiency (EE%) was calculated using the following equation:$$\mathrm{EE}\;\left(\%\right)=\frac{\mathrm{DOX}\;\mathrm{input}-\mathrm{Total}\;\mathrm{amount}\;\mathrm{of}\;\mathrm{DOX}\;\mathrm{in}\;\mathrm{rinse}\;\mathrm{solution}}{\mathrm{DOX}\;\mathrm{input}}\times 100\%$$

#### 2.2.5. Drug Release Studies

The in vitro release experiments of DOX-loaded peptide hydrogels (15 mg/mL) were carried out under different pH conditions (pH 5.8 and 7.4). After the gels were stabilized, 1000 μL phosphate buffer saline was carefully added to the surface of the hydrogels. The release experiments were carried out in a thermostatic shaker (LY20-211C, Wuhan, China) at 37 ± 0.5 °C and 100 rpm for 120 h. At predesigned intervals, the entire volume of release medium above the hydrogels was collected for ultraviolet spectrophotometry analysis. Meanwhile, an equal volume of fresh release medium was added immediately. The in vitro DOX release profile was finished by plotting the percentage cumulative drug release against time.

#### 2.2.6. Transmission Electron Microscopy (TEM)

The micromorphology of the peptides was characterized via TEM. P1 peptide hydrogels of 15 mg/mL with different pH conditions (pH 5.8 and 7.4) were prepared using the above method and diluted 10 times to 1.5 mg/mL with ultrapure water. Then, several drops of the samples were placed on a copper grid coated with a carbon film and excess liquid removed using filter paper after 5 min. We used 2% phosphotungstic acid solution to stain the samples for 1 min and air-dried them for 24 h at room temperature. Images of the fibril nanostructure were acquired using a Hitachi HT-7700 transmission electron microscope.

#### 2.2.7. Zeta Potential

Zeta potential measurements were performed on a Malvern Zetasizer Nano ZS90 using 1 mg/mL peptide solutions (pH 5.8 and 7.4) in ultrapure water at 25 °C. Two samples were measured with at least three runs per sample.

#### 2.2.8. Thioflavin T (ThT) Assay

A precise weight of 1.595 mg of thioflavin T was dissolved in 1 mL deionized water to obtain a stock solution of ThT. An amount of 1 mL of 1 mg/mL peptide solution was added to 10 μL of ThT stock solution. Then, the mixture was left in the dark for 4 h for complete binding. The fluorescence spectrum was determined with the excitation wavelength at 442 nm, the excitation time at 0.5 s, the emission wavelength between 400 and 550 nm, and the slits at 3 nm.

#### 2.2.9. Circular Dichroism Spectroscopy

The secondary structure of the P1 hydrogel was determined via circular dichroism (CD) using a Jasco J-810 spectropolarimeter at 25 °C. P1 peptide hydrogels of 15 mg/mL under different pH conditions (pH 5.8 and 7.4) were prepared as described above and diluted to 0.5 mg/mL with ultrapure water. The CD spectra from 190 to 260 nm were collected using a path length of 1.0 mm at a speed of 50 nm/min with a response time of 1.0 s.

#### 2.2.10. Rheological Measurements

Rheological measurements were conducted using a HAAKE Mars 40 rotational rheometer. Dynamic frequency and strain sweep tests were performed at 37 °C to simulate human body temperature. The dynamic frequency sweep measurements were performed at frequencies between 1 and 100 rad/s and at a constant strain (1%). Similarly, the dynamic strain sweep measurements were performed with strain between 0.1% and 100% and at a constant frequency (6.28 rad/s). The circle sweep measurements were initially obtained at 1% strain for 120 s, then, at 50% strain for 120 s, and finally, 1% strain for 120 s. 

#### 2.2.11. In Vitro Cytotoxicity Evaluation 

##### Cytotoxicity of the Blank Peptide Hydrogel

Normal NIH3T3 fibroblast cells were used to measure the cytotoxicity of the P1 blank peptide hydrogel via the MTT assay. Briefly, the cells were seeded into a 96-well plate with 50,000 cells per well and cultured in DMEM with 10% fetal bovine serum (FBS) for 24 h. Several different concentrations of blank peptide hydrogel (100 μL/well) were added to each well, and the complete medium was set as the control group. After incubation for 48 h, the supernatant was discarded and 20 μL of MTT solution (5 mg/mL) was added to each well for 4 h. Before the test, the supernatant was replaced with 100 μL DMSO to dissolve crystals by shaking for 10 min. The absorbance value of each supernatant’s optical density (OD) was measured using a microplate reader at 570 nm according to the following equation:$$\mathrm{cell}\;\mathrm{viability}\;\left(\%\right)=\frac{\mathrm{absorbance}\;(\mathrm{test}\;\mathrm{group})}{\mathrm{absorbance}\;(\mathrm{control}\;\mathrm{group})}\times 100\%$$

##### In Vitro Antitumor Efficacy of the DOX-P1 Peptide Hydrogel

The antitumor effect of the DOX-P1 peptide hydrogel was also investigated using the MTT assay. K12 osteosarcoma cells were seeded into a 96-well plate (5 × 104 cells/mL). After incubation for 24 h, the cells were treated with various DOX concentrations of free DOX solution and DOX-P1 peptide hydrogel for a further 48 h. Similarly, the group with added medium served as a control. Subsequently, the supernatant was discarded. The cell viability and the IC50 values were also calculated.

#### 2.2.12. In Vivo Antitumor Studies

To establish a subcutaneous tumor model, K12 cells were inoculated into the right axilla of BALB/c mice (female, 16–20 g). When the tumor volume reached approximately 100 mm^3^, 15 tumor-bearing mice were randomly divided into three groups (n = 5). The mice were intratumorally injected under the different regimens at an equivalent DOX dosage of 10 mg/mL, including saline (200 μL/20 g: the blank group), free DOX (200 μL/20 g), and DOX-P1 peptide hydrogel (200 μL/20 g). Tumor volume and body weight were measured every day (tumor volume = 0.5 × length × width^2^) (mm^2^). The mice were sacrificed on the seventh day, and the tumors were dissected, photographed, and weighed. Then, the tumor sections from different treatment groups were fixed in paraformaldehyde (4 wt.%) and analyzed using the H&E staining and TUNEL assays.

#### 2.2.13. In Vivo Biodistribution of the Peptide Hydrogel

In vivo biodistribution of the peptide hydrogel was performed using a near-infrared fluorescent dye in an in vivo imaging system. DiR as a model drug was dissolved in PBS to study the biodistribution in K12 tumor-bearing mice. When tumor volume reached 200 mm^3^, 12 mice were divided into two groups, including the free DiR group (the control group) and the DiR-P1 group (the DiR-P1 peptide hydrogel group). An amount of 100 μL of free DiR and DiR-P1 (200 μg/mL of DiR) was intratumorally injected into the tumor-bearing mice. At predesignated time points (1, 6, 12, 24, 48, and 96 h), the mice were anesthetized and imaged by a living imaging system (λ_Ex_ = 748 nm, λ_Em_ = 780 nm). The mice were then sacrificed after 96 h, and the major organs (heart, liver, spleen, lung, kidney) and tumor were separated for further biodistribution analysis. 

#### 2.2.14. Statistical Analysis

For statistical analysis, we utilized SPSS Prism Software17.0 (IBM, Chicago, IL, USA). All data are presented as mean ± standard deviation (SD). A Student’s *t*-test and one-way analysis of variance (ANOVA) were carried out to compare the values of multigroups. A *p*-value of less than 0.05 was considered statistically significant.

## 3. Results and Discussion

### 3.1. Peptide Design, Synthesis, and Purification

The novel sequence, Ac-FFFGSLKGK (P1), with strong self-assembly capability was proposed as a surfactant-like peptide. The details of the design are as follows: (1) A consecutive-block FFF (Phe-Phe-Phe) as the hydrophobic tail can self-assemble at low concentrations [24]. (2) An acetyl group (Ac-) is conjugated with the N-terminus to reduce the total charge and solubility of peptides. (3) A charged-block KGK (Lys-Gly-Lys) is placed at the C-terminus, which resembles a hydrophilic head, in response to pH changes. (4) The Gly-Ser-Leu motif in the middle is utilized to improve performance and stability. At the same time, we also designed Ac-FFFGSLKG (P0) and Ac-FFFGSLKGD (P2) for comparison (Table 1). In this design, a polar amino acid was inserted at the end of the sequence to enhance electrostatic repulsion and maintain system dispersion. Compared with P1, P0 lacks a polar amino acid (lysine, K, PI = 9.74) at the end, while the lysine at the end of P1 is replaced with an aspartic acid (D, PI = 2.97) in P2. They were successfully synthesized via the solid-phase synthesis method. The purity levels of the peptide powders were all above 97%, as confirmed by high performance liquid chromatography (Appendix A). The mass spectrometry confirmation of P1 is illustrated in Appendix A.

### 3.2. Self-Assembly and Gelation of the Peptide Hydrogels

We evaluated the self-assembly and gelation of these peptides at different pH values. The self-assembly of the designed peptides was promoted by adjusting pH with the addition of NaOH or HCl in water. P1 appeared as a fluid liquid in acidic conditions, even at a high concentration of 20 mg/mL. In contrast, a stable hydrogel could be formed under neutral conditions (pH 7.4) of 10 mg/mL and remained stable under alkaline conditions (pH 9.2), which could make it a potential pH-sensitive carrier for anticancer agents. P0 and P2 could both form hydrogels at pH 5.8 and 7.4, but their gel states were unstable at pH 7.4 (Appendix A). The reasons for this may be that the electrostatic repulsion of P0 and P2 was dominant in neutral conditions, leading to hydrogel instability. Compared with the hydrogels formed by P1, the P0 hydrogel dehydrated under a certain external force and could not return to its initial state. This result further indicated that a basic amino acid in the terminal of the P0 peptide might increase its water retention capacity. In addition, the rate of gelation increased with a higher concentration of the P1 peptide. According to the results shown in Appendix A, we chose 15 mg/mL of P1 as the optimal concentration.

### 3.3. PH Responsiveness of Peptide Hydrogels

We next assessed the in vitro release behavior of DOX from DOX-loaded peptide hydrogels in phosphate buffer saline solutions (pH 5.8 and 7.4) at 37 °C. As shown in Figure 2A, DOX was rapidly released from the DOX-P1 peptide hydrogel at the beginning of 15 h. Then, the release rate of DOX gradually decreased. More importantly, DOX displayed sustained and selective release in the tumor regions, with cumulative release percentages of 10.09% (pH 7.4) and 36.4% (pH 5.8) over 120 h. This result suggested that the DOX-P1 hydrogel had significantly higher sensitivity to the acidic conditions of the tumor microenvironment (pH 5.8 versus 7.4, up to 3.6-fold) than the DOX-P2 peptide hydrogel (Appendix A). The encapsulation efficiency of P1 was 99.790%, while that of P2 was only 88.225% (Table 2). This result implied that the interaction between P2 and DOX hindered the stability of nanofiber networks. Taken together, the DOX-P1 hydrogel could serve as a pharmaceutical reservoir for the responsive and sustained release of DOX, so we chose the P1 peptide for the following experiments.

### 3.4. Characterization of the P1 Peptide Hydrogel

#### 3.4.1. Transmission Electron Microscopy

To seek a further understanding of the self-assembly behavior of the P1 peptide hydrogel, transmission electron microscopy (TEM) was used to characterize its structural morphology in its assembled state. The TEM image shown in Figure 2B reveals long and curved nanofiber networks of the P1 peptide at pH 7.4, indicative of a right-handed helix structure. In contrast, the nanofibers became incomplete in acidic conditions, indicating nanofiber breakage of the hydrogel. We were also curious about whether the presence of DOX would disrupt hydrogel morphology. As shown in Appendix A, we found that DOX could be effectively encapsulated into the P1 hydrogel without influencing the formation of the peptide hydrogel. DOX was mainly adsorbed on the surface of nanofibers through noncovalent interactions, which is consistent with previous findings [25].

#### 3.4.2. Secondary Structure 

The corresponding CD spectrum displayed a negative peak at 190 nm and a weak positive peak in the range of 210–230 nm at pH 7.4 (Figure 2C), indicating that P1 peptide hydrogels might have multiple secondary structures, including β-sheet and random coil conformation. A similar phenomenon was also observed in Ac-KEFFFFKE-NH_2_, which is probably attributed to the interference of consecutive phenylalanine to the CD spectrum [26]. To confirm the existence of β-sheets, fluorescence emission spectrum measurements were performed. The maximum emission wavelength of Thioflavin T was red-shifted to 482 nm with a significant fluorescence enhancement (Figure 2D). This result was in tandem with the results determined by CD, and more importantly, it confirmed the existence of β-sheets. The reason may be that the presence of π–π interactions among three phenylalanine residues could promote a higher β-sheet tendency of the P1 peptide hydrogel.

#### 3.4.3. Zeta Potential 

As shown in Figure 2E, the P1 hydrogel had opposite charges at pH 5.8 and 7.4. The hydrogel showed negative charges under neutral conditions, which were especially suitable for encapsulating positively charged doxorubicin. However, while the pH dropped to 5.8, the degree of protonation of basic amino acids in the polypeptide increased, and the electrostatic repulsion of the whole system increased, leading to the collapse of the hydrogel and the release of DOX. It is reasonable to assume that the transition from electrostatic attraction to electrostatic repulsion induced by pH was the reason for the responsiveness release of the drug-loaded hydrogel. 

#### 3.4.4. Rheological Studies 

Eventually, to test the mechanical behavior of the P1 peptide hydrogel, we also performed rheological studies via dynamic strain and frequency scanning measurements. Dynamic strain scanning was used to investigate the linear viscoelastic region of hydrogels. The storage modulus (G′) and the loss modulus (G″) decreased drastically when the strain exceeded 10% (Figure 3A). The dynamic frequency scanning results showed that the G′ of the hydrogel was greater than the G″, and the result was frequency independent (Figure 3B), suggesting that the P1 hydrogel was in a solidlike state with good viscoelasticity. The circle sweep measurements showed that the peptide became shear thinning under high strain and recovered quickly to form a stable hydrogel under low strain (Figure 3C), which satisfied the requirements of injection. These results proved that the DOX-P1 peptide hydrogel could be an injectable carrier material with high pH responsiveness.

### 3.5. In Vitro Cytotoxicity Evaluation

#### 3.5.1. Cytotoxicity of the Blank Peptide Hydrogel

To test the biocompatibility of the blank intratumoral injection hydrogels, we conducted in vitro cell experiments via MTT analysis. The results showed that the survival rates of NIH3T3 cells were above 90% in all the concentration groups after incubation for 48 h (Figure 3D), which means that the blank hydrogel had excellent biocompatibility.

#### 3.5.2. In Vitro Antitumor Efficacy of the DOX-P1 Peptide Hydrogel

Next, we evaluated the cytotoxicity of drug-loaded peptide hydrogels with different concentrations via the inhibition effect on K12 cell proliferation. As shown in Figure 3E, it was clear that both the free DOX solution and the DOX-P1 hydrogel took on concentration dependence. At low concentrations (<5 μg/mL), the survival rates of the drug-loaded hydrogel group were lower than those of the free DOX solution group. A reasonable explanation could be that the cell internalization of nanofibers in the medium increased the cellular uptake of DOX adsorbed on the surface of nanofibers [27,28]. The cell viability of the two groups was similar at concentrations of 5–20 μg/mL. However, the cytotoxicity of the hydrogel group was slightly inferior to that of the free DOX solution group on account of the incomplete release of DOX from the DOX-loaded hydrogel within a short culture time. As shown in Table 3, the IC50 value of the DOX-P1 hydrogel was 2.353 μg/mL, in contrast with 2.989 μg/mL in the free DOX group, suggesting that the encapsulation of DOX into hydrogels could improve the antitumor efficacy. Overall, these results proved that DOX could effectively disassociate from the hydrogel and release part of the free DOX after internalization to inhibit tumor growth.

### 3.6. In Vivo Antitumor Studies of the DOX-P1 Hydrogel

The therapeutic efficiency of the DOX-P1 hydrogel was evaluated in a K12 osteosarcoma orthotopically grown in NOD/SCID mice. We first measured the body weights and tumor volumes of the mice. The body weights of the mice in the free DOX group decreased (Figure 4A), implying toxic side effects of DOX. In comparison with the blank group, the bodyweight of the DOX-P1 hydrogel group showed the same tendency to slightly increase. There were no significant differences between the two groups. These results demonstrated that the encapsulation of DOX into hydrogels could reduce toxic side effects. As shown in Figure 4B, the tumor volume of the blank group rapidly increased from 100 to 800 mm^3^, while tumor growth was significantly inhibited in the free DOX group, and the tumor inhibition rate was 52.36%. Most notably, the tumor inhibition rate was 68.11% in the DOX-P1 hydrogel group, indicating that the DOX-P1 hydrogel had a stronger and more sustained antitumor effect. Next, we assessed the therapeutic efficacy via immunohistochemical analysis (Figure 4D). Hematoxylin and eosin (H&E) staining revealed more karyorrhexis and pyknosis of cellular necrosis in sharp contrast to the blank group, and the tumors treated with the DOX-P1 hydrogel exhibited the most cell death. Compared with the group treated with DOX only, the TUNEL assays displayed stronger green fluorescence-tagged apoptotic cells in the DOX-P1 hydrogel group, which was consistent with the results of the animal experiments. In contrast, there was no green fluorescence in the blank group. These experiments proved that our DOX-P1 hydrogel could promote DOX release in tumor tissues and had superior antitumor efficacy. 

### 3.7. In Vivo Distribution and Intratumoral Retention of the DOX-P1 Hydrogel

To further evaluate intratumoral retention and distribution in the tumor and normal organs, we conducted live animal imaging experiments by imaging K12 tumor-bearing mice. After intratumoral injection of the DiR-P1 hydrogel and free DiR in the two groups, the fluorescence of the DiR-P1 group was concentrated in the tumor regions, and no attenuation was observed at 96 h (Figure 5A). In sharp contrast, the fluorescence intensity and area were much smaller in the free DiR group, which confirmed that the hydrogel showed longer intratumoral retention than in the control group. The same results are shown in Figure 5B, with significantly higher fluorescence intensity in tumors treated with DiR-P1 than in those treated with free DiR. The accumulation of DiR in the heart, spleen, kidney, and liver in the DiR-P1 group was lower than that in the free DiR group. These results demonstrated that the intratumoral injection of P1 hydrogels could significantly enhance drug retention at the tumor site and, at the same time, reduce off-site distribution. 

## 4. Conclusions

A pH-responsive and injectable nonapeptide hydrogel carrying doxorubicin was constructed in this study as a local chemotherapy reservoir. The results showed that the structure of the P1 peptide resembled the surfactant-like peptide, and the hydrophobic interaction between hydrophobic tails and the electrostatic interaction between hydrophilic heads promoted its self-assembly in neutral conditions. Importantly, the P1 hydrogel presented high drug encapsulation efficiency and injectability and high sensitivity to acidic conditions (pH 5.8 versus 7.4, up to 3.6-fold). Furthermore, in vivo and in vitro studies also demonstrated that the encapsulation of DOX into P1 hydrogels not only amplified the therapeutic effect but also significantly increased DOX accumulation at the tumor sites. We believe that the P1 hydrogel with tumor-targeted ability could be a promising approach for local osteosarcoma chemotherapy.

## Figures and Tables

**Figure 1 pharmaceutics-15-00668-f001:**
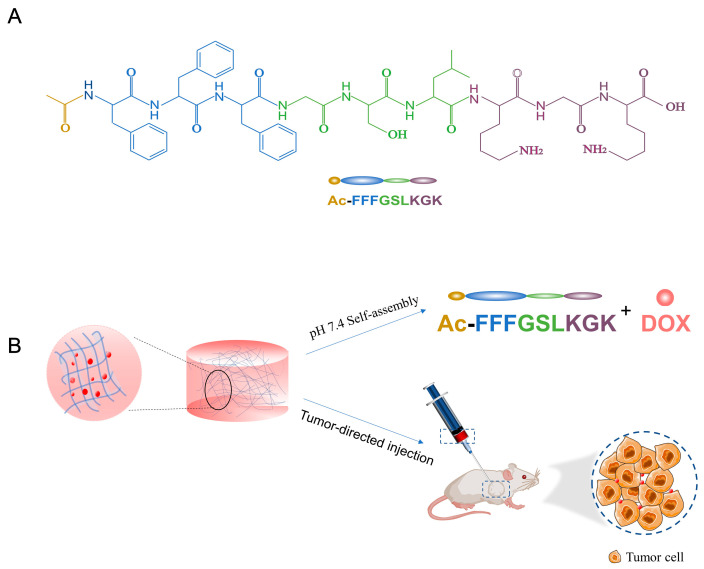
Schematic illustration of pH-responsive and injectable DOX-loaded hydrogels as drug delivery systems for carcinoma therapy. (**A**) Molecular structure of Ac-FFFGSLKGK. (**B**) The peptide can self-assemble into hydrogels in neutral conditions and effectively encapsulate DOX. After intratumoral injection, the DOX from DOX-loaded hydrogels was selectively released into the tumor regions.

**Figure 2 pharmaceutics-15-00668-f002:**
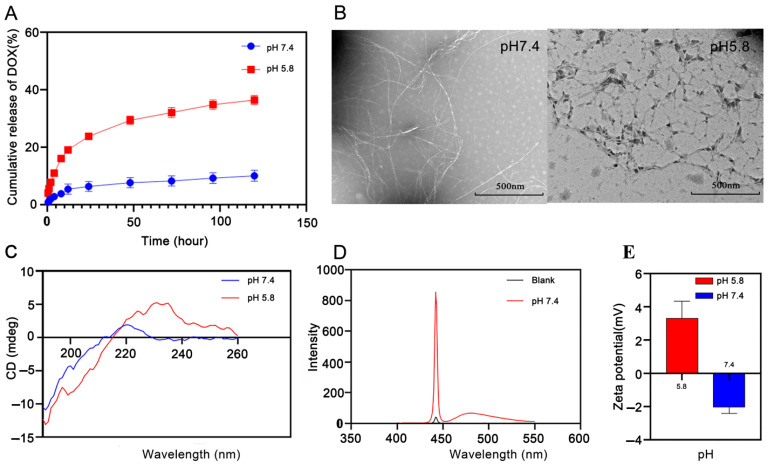
(**A**) Cumulative release of DOX from DOX-P1 hydrogel. (**B**) Transmission electron microscopy of 0.15 wt.% P1 hydrogel under pH 7.4 and 5.8. (**C**) CD spectrum of P1 nonapeptide under pH 7.4 and 5.8. (**D**) ThT assay of P1 nonapeptide under pH 7.4 and 5.8. (**E**) Zeta potential of P1 nonapeptide under pH 7.4 and 5.8 (n = 3).

**Figure 3 pharmaceutics-15-00668-f003:**
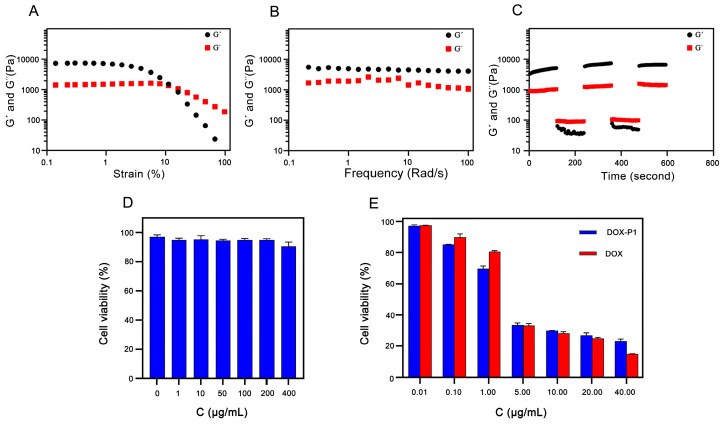
(**A**) The dynamic strain sweep of P1 hydrogels. The frequency was 6.28 rad/s. (**B**) The dynamic frequency sweep of P1 hydrogels. (**C**) The circle sweep of P1 hydrogels. (**D**) Cell viability of NIH3T3 cells at different concentrations of the blank peptide hydrogel (n = 3). (**E**) Cell viability of 4T1 cells coincubated with DOX or DOX-P1 hydrogel (n = 3).

**Figure 4 pharmaceutics-15-00668-f004:**
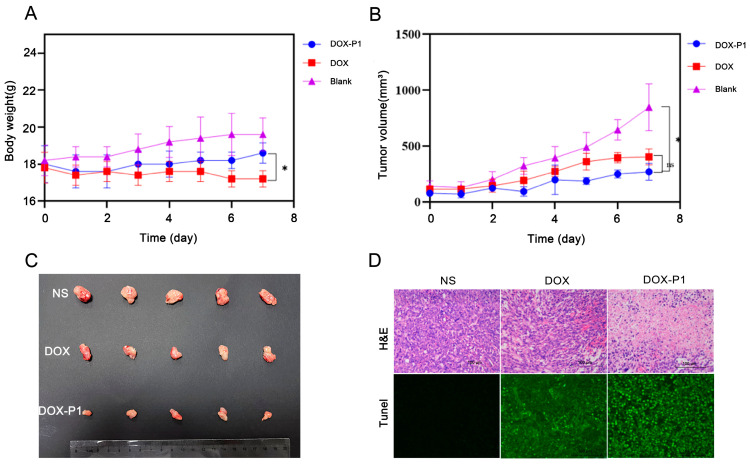
(**A**) The body weight change curve of mice after administration. (**B**) The tumor volume change curve of mice after administration. (**C**) Photographs of the tumors harvested from the mice in the blank group (NS), the DOX group (DOX), and the DOX-P1 group (DOX-P1) 7 d after administration (n = 5). (**D**) H&E staining (top) and TUNEL assays (bottom) of tumor sections after different treatments within 7 d. NS—normal saline; DOX—doxorubicin; DOX-P1—doxorubicin-loaded P1 nonapeptide hydrogel (one-way ANOVA, mean ± SD, * *p* < 0.05).

**Figure 5 pharmaceutics-15-00668-f005:**
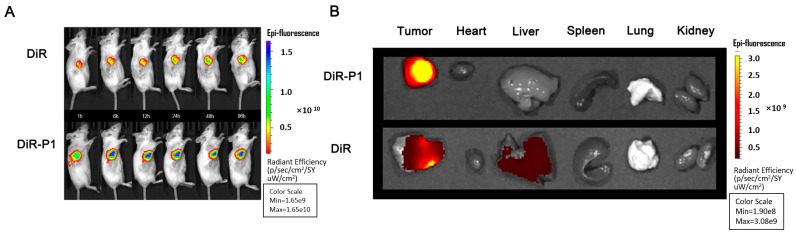
(**A**) In vivo fluorescence imaging of mice over time after administration. (**B**) Fluorescence imaging of tumors, hearts, livers, spleens, lungs, and kidneys of mice.

**Table 1 pharmaceutics-15-00668-t001:** Sequence of designed peptides.

Peptide	Sequence
P0	Ac-FFFGSLKG
P1	Ac-FFFGSLKGK
P2	Ac-FFFGSLKGD

**Table 2 pharmaceutics-15-00668-t002:** Efficiency of drug encapsulation of nonapeptides (n = 6).

Peptide	P1	P2
Encapsulation efficiency (%)	99.790	88.225%

**Table 3 pharmaceutics-15-00668-t003:** IC50 of DOX and DOX-P1 (n = 3).

Group	IC50 (μg/mL)	SD	*p*
DOX	2.989 μg/mL	0.193	0.0239 *
DOX-P1	2.353 μg/mL	0.244	

*p*: DOX group vs. DOX-P1 group; * 0.01 < *p* < 0.05.

## Data Availability

All data relevant to the publication are included.

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
