# Peer review of "Sustained and Targeted Delivery of Self-Assembled Doxorubicin Nonapeptides Using pH-Responsive Hydrogels for Osteosarcoma Chemotherapy"

_pharmaceutics, 2023, doi:10.3390/pharmaceutics15020668_

Round 1

Reviewer 1 Report

In this manuscript, the authors developed a nonapeitide, Ac-FFFGSLKGK, that forms a pH-responsive hydrogel. The hydrogel loaded with anticancer agents doxorubicin (DOX) is stable at pH 7.4 and release DOX at 5.8. The physicochemical properties of the hydrogel were well characterized by encapsulation capacity, TEM, CD, zeta potential, rheological analysis, and drug release studies. In vitro and in vivo cytotoxicity and tumor inhibitory efficacy experiments were also carried out. The results suggest that the reported hydrogel is a promising candidate for targeted chemotherapy. This reviewer recommends publication of this manuscript in Pharmaceuticals after minor revision.

Abstract: the abbreviation of transmission electron microscopy is TEM, not TME. 

Author Response

Response to the reviewers and comments

Thanks for your detailed and constructive advice on our article entitled “Sustained and targeted delivery of self-assembled doxorubicin nonapeptides by pH-responsive hydrogels for osteosarcoma chemotherapy”. We respect the comments of the reviewers and express our great appreciation to you for your careful assessment and great effort paid on our research. We promise to carefully revised the manuscript to reach the high level of Pharmaceutics. In detail, the specific questions and concerns raised by you and reviewers (italicized) are addressed point by point. And corresponding changes in the manuscript was labeled as yellow in background color. Please also be noted that the revised manuscript has been extensively polished by language editing service offered by MDPI. Thank you for considering of the publishment of our work.

Sincerely,

Yuan Zhang M.D., Director

Department of Orthopedics

Xinqiao Hospital, Army Medical University

Chonging, 400038, China.

zhangyuan@tmmu.edu.cn

joint_chueng@hotmail.com

Reviewer 3

In this manuscript, the authors developed a nonapeitide, Ac-FFFGSLKGK, that forms a pH-responsive hydrogel. The hydrogel loaded with anticancer agent doxorubicin (DOX) is stable at pH 7.4 and release DOX at 5.8. The physicochemical properties of the hydrogel were well characterized by encapsulation capacity, TEM, CD, zeta potential, rheological analysis, and drug release studies. In vitro and in vivo cytotoxicity and tumor inhibitory efficacy experiments were also carried out. The results suggest that the reported hydrogel is a promising candidate for targeted chemotherapy. This reviewer recommends publication of this manuscript in Pharmaceuticals after minor revision.

Abstract: the abbreviation of transmission electron microscopy is TEM, not TME.

Reply: Thank you for your positive comments on our work. And we have corrected this mistake in the revised abstract.

Please let me know for any of your question.

Sincerely,

Yuan Zhang M.D., Director

Department of Orthopedics

Xinqiao Hospital, Army Medical University

Chonging, 400038, China.

zhangyuan@tmmu.edu.cn

joint_chueng@hotmail.com

Reviewer 2 Report

The manuscript entitled “Sustained and targeted delivery of self-assembled doxorubicin nonapeptides by pH-responsive hydrogels for osteosarcoma chemotherapy” deals with an important and interesting topic. In my opinion it contains quality chosen techniques for hydrogels design and their characterization as well as determination of DOX-P1 complex. However, I have some prepositions in order to improve the manuscript, they are listed below:

In Methods section it is necessary to clarify:

Why were the rheological experiments done at 37°C?

Why was the circle sweep test done? I assume that it was also made at a constant frequency, which needs to be emphasized. This test is not usual during rheological measurements, so it is necessary to explain what is expected to be obtained with this test.

In Results and discussion section it is necessary to clarify:

What was the selected value of deformation based on the figure 3A, that was used in frequency sweep tests?

What does it mean when you claim that P1 hydrogel is with good viscoelasticity?

How would you explain the claim based on Figure 3C that “peptide became shear-thinning under high strain and recovered quickly to a stable hydrogel under low strain”?

Please explain how you came to the following conclusion based on the measurement of the zeta potential at two pH values: “It was reasonable to assume that the transition from the electrostatic attraction to electrostatic repulsion induced by pH was the reason for the responsiveness release of the drug-loaded hydrogel”.

Author Response

Response to the reviewers and comments

Thanks for your detailed and constructive advice on our article entitled “Sustained and targeted delivery of self-assembled doxorubicin nonapeptides by pH-responsive hydrogels for osteosarcoma chemotherapy”. We respect the comments of the reviewers and express our great appreciation to you for your careful assessment and great effort paid on our research. We promise to carefully revised the manuscript to reach the high level of Pharmaceutics. In detail, the specific questions and concerns raised by you and reviewers (italicized) are addressed point by point. And corresponding changes in the manuscript was labeled as yellow in background color. Please also be noted that the revised manuscript has been extensively polished by language editing service offered by MDPI. Thank you for considering of the publishment of our work.

Sincerely,

Yuan Zhang M.D., Director

Department of Orthopedics

Xinqiao Hospital, Army Medical University

Chonging, 400038, China.

zhangyuan@tmmu.edu.cn

joint_chueng@hotmail.com

Reviewer 1

The manuscript entitled “Sustained and targeted delivery of self-assembled doxorubicin nonapeptides by pH-responsive hydrogels for osteosarcoma chemotherapy” deals with an important and interesting topic. In my opinion it contains quality chosen techniques for hydrogels design and their characterization as well as determination of DOX-P1 complex. However, I have some prepositions in order to improve the manuscript, they are listed below:

In Methods section it is necessary to clarify: Why were the rheological experiments done at 37°C?

Reply: Thank you for this question. We performed the rheological experiments at 37°C as we need to make sure that the hydrogel appears solid form in the patients’ body. 37°C was used to simulate human body temperature.

Why was the circle sweep test done? I assume that it was also made at a constant frequency, which needs to be emphasized. This test is not usual during rheological measurements, so it is necessary to explain what is expected to be obtained with this test.

Reply: We did the sweep test at the beginning at a constant frequency because we tried to observe the trend of G 'and G' 'with the change of strain. Therefore, we can have a preliminary understanding of the mechanical properties of the hydrogel.

In Results and discussion section it is necessary to clarify:

What was the selected value of deformation based on the figure 3A, that was used in frequency sweep tests?

What does it mean when you claim that P1 hydrogel is with good viscoelasticity?

How would you explain the claim based on Figure 3C that “peptide became shear-thinning under high strain and recovered quickly to a stable hydrogel under low strain”?

Reply: The frequency of frequency sweep tests was 6.28 rad/s and we have added it in the figure legend. When the strain was below 10%, G’ was higher than G’’, the hydrogel appeared solid state. When the strain was above 10%, G’ was lower than G’’, the hydrogel appeared liquid state so that the hydrogel has good viscoelasticity. In Fig.3C, when the strain was 1%, G’ was higher than G’’. When the strain changed to 100%, G’ became lower than G’’ and when the strain changed to 100% again, the G’ became higher than G’’ immediately so proposed that “peptide became shear-thinning under high strain and recovered quickly to a stable hydrogel under low strain”.

Please explain how you came to the following conclusion based on the measurement of the zeta potential at two pH values: “It was reasonable to assume that the transition from the electrostatic attraction to electrostatic repulsion induced by pH was the reason for the responsiveness release of the drug-loaded hydrogel”.

Reply: When the pH was 7.4, Dox appears positive-charged. Meanwhile, the zeta potential of P1peptide was negative. The electrostatic attraction helps the hydrogel formation. However, while the pH drops to 5.8, The degree of protonation of basic amino acids in the polypeptide increases, and the electrostatic repulsion of the whole system increases, leading to the collapse of hydrogel and the release of DOX.

Reviewer 3 Report

The present manuscript describes the synthesis of a pH-responsive nonapeptide as a local delivery system of doxorubicin for the treatment of osteosarcoma. The manuscript would significantly benefit from proofreading by a native English speaker.

Introduction: please correct grammar and syntax errors throughout the text e.g.

‘and exhibited aggressive progress and invasion property’, ‘Hydrogels are three-dimensional (3D) porous networks with high water contents by physical or chemical crosslinking[8], which have been envisioned as potentially good can-didates in biomedical fields.’, ‘the pH of the extracellular environment in solid tumors was slightly acidic’, ‘drug-loaded capacity’, ‘Utilizing Wang resin for solid carriers was because of the carboxyl-terminal of P1.’, ‘the complete medium was served as a control’, ‘P0 hydrogel occurred syneresis under certain external mechanical’, ‘The dynamic frequency scanning showed that the hydrogel had certainly independent of frequency’, ‘As shown in Fig.4B, the rapid tumor growth of the blank group was found in mice, increasing the volume from 100 mm3 to 800 mm3.’

Please provide references for the following statement: ‘In this work, consecutive block FFF and charged block KGK structurally mimic the hydrophobic tail and hydrophilic head of SLPs, which are used to modulate molecular assembly and hydrogel formation.’

What do authors mean by performance? In what aspect can this sequence improve the performance of the peptide? ‘Gly-Ser-Leu motif in the middle, which is a part of an antimicrobial peptide[22], is used to improve its performance’

2.2.2. Preparation of Hydrogels: what is the concentration of the peptide used to prepare the hydrogel? why did authors dissolve the peptide in water and then adjusted the pH to 7.4 instead of directly dissolving the peptide in e.g. PBS pH 7.4? 

2.2.3. Gelation Behavior at Different pH Values: why weren’t buffered solutions of different pH values used directly instead?

2.2.5. Drug Release Studies: since the hydrogels are biodegradable and unstable at acidic conditions, is there any contribution from the peptide released in the medium to DOX absorbance? How was that confirmed?

2.2.6. Transmission Electron Microscopy (TEM). ‘..after immersed the grids for 5 min. ‘ In what where the grids immersed in?

2.2.10. Rheological Measurements: please correct ‘gap size of 5 nm.’

2.2.12. In Vivo Anti-Tumor Studies: was the peptide hydrogel already preformed when administered?

Figure S2. At which value of the y axis does the [M+1H] correspond to? Is this [M+2H] a fragment of the peptide? Remove Figure S2 form the manuscript, since you already have it in the SI. Same for figure S3, S4.

Figure 2: Improve the quality of all graphs.

Please rephrase: ‘DOX-P1 peptide hydrogel showed a high-speed release of DOX’

3.4.1. Transmission Electron Microscopy: ‘DOX was mainly adsorbed on the surface of nanofibers by non-covalent interaction, which was consistent with the previous findings’ how can authors support this claim? Confocal microscopy of the drug loaded hydrogel should be performed to confirm DOX distribution.

Graphs should be better organized with respect to the text. They should follow and not precede the text.

Add the SD value for the IC50 to confirm statistical significance between the DOX and DOX-P1 groups.

How is ‘cell internalization of nanofibers’ confirmed in the study?

3.6. In Vivo Anti-tumor Studies of DOX-P1 Hydrogel: K12 osteosarcoma was not orthotopically grown in BALB/c mice. Authors should correct this inconsistency.

Author Response

(The authors gave the same response as above.)

Round 2

Reviewer 2 Report

In my opinion authors made an effort to imrove the Manuscript. 

Author Response

Thank you for the comments.

Reviewer 3 Report

A point-by point response to reviewer's comments is missing. From the highlighted changes in the revised manuscript it is evident that authors have only addressed reviewer's points related to grammatical/syntax errors. Please provide a point-by-point response to all reviewer's comments. 

Author Response

Thanks for your detailed and constructive advice on our article entitled “Sustained and targeted delivery of self-assembled doxorubicin nonapeptides by pH-responsive hydrogels for osteosarcoma chemotherapy”. We respect the comments of the reviewers and express our great appreciation to you for your careful assessment and great effort paid on our research. We promise to carefully revised the manuscript to reach the high level of Pharmaceutics. In detail, the specific questions and concerns raised by you and reviewers (italicized) are addressed point by point. And corresponding changes in the manuscript was labeled as yellow in background color. Please also be noted that the revised manuscript has been extensively polished by language editing service offered by MDPI. Thank you for considering of the publishment of our work.

Sincerely,

Yuan Zhang M.D., Director

Department of Orthopedics

Xinqiao Hospital, Army Medical University

Chonging, 400038, China.

zhangyuan@tmmu.edu.cn

joint_chueng@hotmail.com

Reviewer 2

The present manuscript describes the synthesis of a pH-responsive nonapeptide as a local delivery system of doxorubicin for the treatment of osteosarcoma. The manuscript would significantly benefit from proofreading by a native English speaker.

Reply: Thank you for great suggestion. Please also be noted that the revised manuscript has been extensively polished by language editing service offered by MDPI.

Introduction: please correct grammar and syntax errors throughout the text e.g. ‘and exhibited aggressive progress and invasion property’, ‘Hydrogels are three-dimensional (3D) porous networks with high water contents by physical or chemical crosslinking[8], which have been envisioned as potentially good candidates in biomedical fields.’, ‘the pH of the extracellular environment in solid tumors was slightly acidic’, ‘drug-loaded capacity’, ‘Utilizing Wang resin for solid carriers was because of the carboxyl-terminal of P1.’, ‘the complete medium was served as a control’, ‘P0 hydrogel occurred syneresis under certain external mechanical’, ‘The dynamic frequency scanning showed that the hydrogel had certainly independent of frequency’, ‘As shown in Fig.4B, the rapid tumor growth of the blank group was found in mice, increasing the volume from 100 mm3 to 800 mm3.’

Reply: Sorry for these flaws. We have corrected grammar and syntax errors mentioned before.

Please provide references for the following statement: ‘In this work, consecutive block FFF and charged block KGK structurally mimic the hydrophobic tail and hydrophilic head of SLPs, which are used to modulate molecular assembly and hydrogel formation.

Reply: Thank you for this kind remind. The reference has been added as: Amphipathic design dictates self-assembly, cytotoxicity and cell uptake of arginine-rich surfactant-like peptides

What do authors mean by performance? In what aspect can this sequence improve the performance of the peptide? ‘Gly-Ser-Leu motif in the middle, which is a part of an antimicrobial peptide[22], is used to improve its performance’

Reply: We are so sorry that the description of the motif is not precise. The function of the motif was not clearly clarified in the reference. We decided to delete the sentence as the motif do not play a significant role in the formation of the hydrogel. The role of FFF and KGK was explained in the article.

2.2.2. Preparation of Hydrogels: what is the concentration of the peptide used to prepare the hydrogel? why did authors dissolve the peptide in water and then adjusted the pH to 7.4 instead of directly dissolving the peptide in e.g. PBS pH 7.4?

2.2.3. Gelation Behavior at Different pH Values: why weren’t buffered solutions of different pH values used directly instead?

Reply to 2.2.2 and 2.2.3: The concentration of the peptide was 15 mg/mL. In the article, we used water as the solvent of the peptide and we did not use PBS solution as the solvent so that we have to adjust the pH value.

2.2.5. Drug Release Studies: since the hydrogels are biodegradable and unstable at acidic conditions, is there any contribution from the peptide released in the medium to DOX absorbance? How was that confirmed?

Reply: The UV/vis spectrum of DOX and the peptide was performed. The maximum absorption peak of DOX was 481nm while the absorption of the peptide at 481nm wan less than 0.2 and we believe that the absorbance of peptide did not influence the detection of DOX release.

2.2.6. Transmission Electron Microscopy (TEM). ‘..after immersed the grids for 5 min. ‘ In what where the grids immersed in?

Reply: The grids was immersed in the 1.5 mg/mL peptide solution with different pH values. The copper grip was placed on the table and the peptide solution was dropped on the surface of the copper grip to stand for 5min. We have changed ‘removed excess liquid by a filter paper after immersed the grids for 5 min.’ to ‘removed excess liquid by a filter paper after 5 min’.

2.2.10. Rheological Measurements: please correct ‘gap size of 5 nm.’

Reply: We have corrected the sentence into the gap size of the plate was 5 nm.

2.2.12. In Vivo Anti-Tumor Studies: was the peptide hydrogel already preformed when administered?

Reply: We prepared the peptide hydrogel at least 12h before administration to ensure its stability.

Figure S2. At which value of the y axis does the [M+1H] correspond to? Is this [M+2H] a fragment of the peptide? Remove Figure S2 form the manuscript, since you already have it in the SI. Same for figure S3, S4.

Reply: The value was marked on the top of the [M+1H] axis not the y axis. The value was 1073.21. [M+2H] is a fragment of the peptide generated in the process of LC-MS. We have removed all the supplementary information from the manuscript.

Figure 2: Improve the quality of all graphs.

Reply: Thank you for this advice. The high-quality image of Fig.2 was provided in the revised work.

Please rephrase: ‘DOX-P1 peptide hydrogel showed a high-speed release of DOX

Reply: We have changed the sentence into DOX was rapidly released from DOX-P1 peptide hydrogel.

3.4.1. Transmission Electron Microscopy: ‘DOX was mainly adsorbed on the surface of nanofibers by non-covalent interaction, which was consistent with the previous findings’ how can authors support this claim? Confocal microscopy of the drug loaded hydrogel should be performed to confirm DOX distribution.

Reply: We have proved the claim in Fig.S4 that DOX was mainly adsorbed on the surface of nanofibers. We think that confocal microscopy is hard to confirm DOX distribution because magnification of confocal microscopy is not enough to observe such a tiny structure. The second reason is performing fluorescent marking on the whole nanofiber is not practical.

Graphs should be better organized with respect to the text. They should follow and not precede the text.

Reply: Thank you for this advice. We have reorganized the graphs in the revised version as you asked.

Add the SD value for the IC50 to confirm statistical significance between the DOX and DOX-P1 groups.

Reply: We have added the SD value for the IC50 in Table.3. Thanks.

How is ‘cell internalization of nanofibers’ confirmed in the study?

Reply: We did not confirm the cell internalization of nanofibers in the study. This is just a speculation according to the references.

3.6. In Vivo Anti-tumor Studies of DOX-P1 Hydrogel: K12 osteosarcoma was not orthotopically grown in BALB/c mice. Authors should correct this inconsistency.

Reply: We made a mistake when writing the article and the strain of the mice was NOD/SCID.

Round 3

Reviewer 3 Report

1.Reply to 2.2.2 and 2.2.3: The concentration of the peptide was 15 mg/mL. In the article, we used water as the solvent of the peptide and we did not use PBS solution as the solvent so that we have to adjust the pH value.

Reviewer’s comment: Authors should reply as to why they followed this approach.  

2. Reply: We have corrected the sentence into the gap size of the plate was 5 nm.

Reviewer’s comment: Do authors mean 5 μm? How is it possible to have a gap size of 5 nm during a rheological measurement?

3.Reply: The value was marked on the top of the [M+1H] axis not the y axis. The value was 1073.21. [M+2H] is a fragment of the peptide generated in the process of LC-MS. We have removed all the supplementary information from the manuscript.

Reviewer’s comment: The question was referring to the concentration of the  [M+2H] fragment, which seems to be much more abundant than the peptide of interest.

4. Reply: We have proved the claim in Fig.S4 that DOX was mainly adsorbed on the surface of nanofibers. We think that confocal microscopy is hard to confirm DOX distribution because magnification of confocal microscopy is not enough to observe such a tiny structure. The second reason is performing fluorescent marking on the whole nanofiber is not practical.

Reviewer’s comment: Confirming the distribution of fluorescent molecules in peptide hydrogels using confocal microscopy has been done in the past. Please check the literature e.g. https://doi.org/10.1021/acs.molpharmaceut.8b01221 Localization of doxorubicin within the peptide hydrogel will be confirmed through the fluorescent signal emitted by DOX and not by localizing the molecule itself.

5. Reply: We made a mistake when writing the article and the strain of the mice was NOD/SCID.

Reviewer’s comment: The question was about the site of tumor implantation. If the animal model was orthotopic then cancer cells should have been implanted in the bone of the animals and not subcutaneously as authors describe in the manuscript.

Author Response

Comments and Suggestions for Authors

1.Reviewer’s comment: Authors should reply as to why they followed this approach.  

Reply: We followed this approach with the concern that other ions in PBS solution would affect the dissolution and gelation of polypeptides, so we use water as the solvent in the whole article.

  1. Reviewer’s comment: Do authors mean 5 μm? How is it possible to have a gap size of 5 nm during a rheological measurement?

Reply: We have deleted the sentence from page 5/13, and the we felt sorry about this mistake.

3.Reviewer’s comment: The question was referring to the concentration of the  [M+2H] fragment, which seems to be much more abundant than the peptide of interest.

Reply: In this mass spectrometry experiment, most molecules are ionized into molecular ion peaks with two charges and this might be determined by the property of the molecule. The y axis of the [M+1H] correspond to the relative abundance compared to the [M+2H].

  1. Reviewer’s comment: Confirming the distribution of fluorescent molecules in peptide hydrogels using confocal microscopy has been done in the past. Please check the literature e.g. https://doi.org/10.1021/acs.molpharmaceut.8b01221Localization of doxorubicin within the peptide hydrogel will be confirmed through the fluorescent signal emitted by DOX and not by localizing the molecule itself.

Reply: Thank you for your recommendation. We have cited this publication to further enhance the comprehension of our work.

Karavasili C, Andreadis DA, Katsamenis OL, Panteris E, Anastasiadou P, Kakazanis Z, Zoumpourlis V, Markopoulou CK, Koutsopoulos S, Vizirianakis IS, Fatouros DG. Synergistic Antitumor Potency of a Self-Assembling Peptide Hydrogel for the Local Co-delivery of Doxorubicin and Curcumin in the Treatment of Head and Neck Cancer. Mol Pharm. 2019, 16, 2326-2341.

  1. Reviewer’s comment: The question was about the site of tumor implantation. If the animal model was orthotopic then cancer cells should have been implanted in the bone of the animals and not subcutaneously as authors describe in the manuscript.

Reply: This is a critical but constructive idea. Yes, we agree that the orthotopic osteosarcoma model is an ideal objective to research the osteosarcoma therapy, and there are several high-quality studies were based on this model, but most of them are focused on the surgical treatment of osteosarcoma. And other disadvantages of orthotopic osteosarcoma model is the low successful rate in tumor implantation and development, and increased difficulty in analyzing the pharmacokinetics and pharmacodynamics in bone tissue. Therefore, we chose the subcutaneous osteosarcoma model to finish the in vivo study, as most of the study in pharmaceutic field did.